# Prognostic Value of Mandard’s Tumor Regression Grade (TRG) in Post Chemo-Radiotherapy Cervical Cancer

**DOI:** 10.3390/diagnostics13203228

**Published:** 2023-10-17

**Authors:** Giulia Scaglione, Damiano Arciuolo, Antonio Travaglino, Angela Santoro, Giuseppe Angelico, Saveria Spadola, Frediano Inzani, Nicoletta D’Alessandris, Antonio Raffone, Caterina Fulgione, Belen Padial Urtueta, Stefania Sfregola, Michele Valente, Francesca Addante, Antonio d’Amati, Federica Cianfrini, Alessia Piermattei, Luigi Pedone Anchora, Giovanni Scambia, Gabriella Ferrandina, Gian Franco Zannoni

**Affiliations:** 1Gynecopathology and Breast Pathology Unit, Department of Woman and Child’s Health and Public Health Sciences, Fondazione Policlinico Universitario Agostino Gemelli IRCCS, 00168 Rome, Italy; giulia.scaglione@policlinicogemelli.it (G.S.); damiano.arciuolo@policlinicogemelli.it (D.A.); antonio.travaglino.ap@gmail.com (A.T.); angela.santoro@policlinicogemelli.it (A.S.); nicoletta.dalessandris@policlinicogemelli.it (N.D.); belen.padialurtueta@guest.policlinicogemelli.it (B.P.U.); stefania.sfregola@guest.policlinicogemelli.it (S.S.); dr.valente.m@gmail.com (M.V.); francesca.addante1@gmail.com (F.A.); federica.cianfrini@policlinicogemelli.it (F.C.); alessia.piermattei@policlinicogemelli.it (A.P.); 2Pathology Unit, Department of Medicine and Technological Innovation, University of Insubria, 21100 Varese, Italy; 3Pathology Unit, Cannizzaro Hospital, 95126 Catania, Italy; giuangel86@hotmail.it (G.A.); saveriaspadola@hotmail.it (S.S.); 4Anatomic Pathology Unit, Department of Molecular Medicine, University of Pavia, 27100 Pavia, Italy; fredianosocrate.inzani@unipv.it; 5Department of Medical and Surgical Sciences (DIMEC), University of Bologna, 40126 Bologna, Italy; anton.raffone@gmail.com; 6Gynecology and Obstetrics Unit, Department of Neuroscience, Reproductive Sciences and Dentistry, Federico II University of Naples, 80131 Naples, Italy; caterina.fulgione@gmail.com; 7Department of Precision and Regenerative Medicine and Ionian Area (DiMePRe-J), University of Bari “Aldo Moro”, 70100 Bari, Italy; antonio.damati@uniba.it; 8Gynecologic Oncology Unit, Department of Woman and Child’s Health and Public Health Sciences, Fondazione Policlinico Universitario Agostino Gemelli IRCCS, 00168 Rome, Italy; luigi.pedoneanchora@policlinicogemelli.it (L.P.A.); giavanni.scambia@policlinicogemelli.it (G.S.); gabriella.ferrandina@policlinicogemelli.it (G.F.)

**Keywords:** locally advanced cervical cancer, Mandard scoring system, prognostic stratification

## Abstract

In locally advanced cervical cancer (LACC), definitive chemo-radiotherapy is the standard treatment, but chemo-radiotherapy followed by surgery could be an alternative choice in selected patients. We enrolled 244 patients affected by LACC and treated with CT-RT followed by surgery in order to assess the prognostic role of the histological response using the Mandard scoring system. Results: A complete pathological response (TRG 0) was observed in 118 patients (48.4%), rare residual cancer cells (TRG2) were found in 49 cases (20.1%), increased number of cancer cells but fibrosis still predominating (TRG3) in 35 cases (14.3%), and 42 (17.2%) were classified as non-responders (TRG4–5). TRG was significantly associated with both OS (*p* < 0.001) and PFS (*p* < 0.001). The survival curves highlighted two main prognostic groups: TRG1-TRG2 and TRG3-TRG4–5. Main responders (TRG1–2) showed a 92% 5-year overall survival (5y-OS) and a 75% 5-year disease free survival (5y-DFS). Minor or no responders showed a 48% 5y-OS and a 39% 5y-DFS. The two-tiered TRG was independently associated with both DFS and OS in Cox regression analysis. Conclusion. We showed that Mandard TRG is an independent prognostic factor in post-CT/RT LACC, with potential benefits in defining post-treatment adjuvant therapy.

## 1. Introduction

Cervical cancer (CC) is the most common gynecological tumor in underdeveloped countries and the fourth neoplasm in women worldwide [1]. Cervical carcinoma represents the second cause of cancer-related death in women aged 20-39 years, with an estimated nine deaths per week in this age group [2]. Cervical cancer is associated with high-risk HPV infection in the vast majority of cases [3,4]. At early stage, cervical cancer shows excellent prognosis with a 5-year survival rate of around 90%. According to ESMO/ESGO guidelines [5], early cervical cancer should be treated with conization or conization and sentinel lymph node dissection—analyzed via ultrastaging or OSNA technique [6].

Locally advanced cervical cancer (LACC) is defined as a tumor ≥ 4 cm in size and classified as stages IB2-IVA according to the 2018 FIGO stage classification [7,8]. The chosen treatment of LACC has been a subject of considerable debate for a long period. Exclusive chemo-radiotherapy (E-CT/RT) represents the standard treatment with a 5-year survival rate of about 60–75% according to the stage [9]. Chemo-radiotherapy (CT/RT) followed by radical surgery could be an alternative with the aim of enhancing the local disease control and reducing toxicity related to the radiation therapy [10]. This approach, as stated in several studies [10,11,12,13,14], achieves a good percentage of pathological complete response, and a very low rate of locoregional failure; indeed, in the phase II ROMA-2 study, which adopted CT/RT with concomitant boost followed by completion surgery, we registered a rate of pathological complete response of 50.5%, and only a 7% rate of 3-year locoregional failure [10,11,12,13,14,15,16]. In this setting, surgical pathology examination remains the gold standard for the assessment of residual disease [17,18,19]. The completion of surgery after CT/RT could provide relevant prognostic parameters. In the literature, the absence of residual disease is associated with better outcomes in terms of disease-free and overall survival. However, only a few studies have a good sample size. Expert pathologists should evaluate morphological alterations related to CT/RT treatment, and the amount of tumor regression should be integrated in the pathology report.

Indeed, as stated in several studies [20,21,22,23,24,25], histopathological assessment of neoplastic response to treatment provides important prognostic information. Different regression systems describe the extent of regressive changes after CH/RT, and mostly refer to the residual tumor compared to the former tumor site. Tumor regression can be defined as the amount of residual tumor, as a percentage, or as the relationship between residual tumor and regressive fibrosis. 

Currently, different scores have been proposed in cervical cancer, but their prognostic value in terms of overall survival (OS) and progression-free survival (PFS) are still undefined [19,26,27,28,29]. 

The five-tiered Mandard tumor regression score (TRG) represents the most frequently used system in different solid cancers, including esophageal and colorectal neoplasms [30]. The TRG refers to the amount of therapy-induced fibrosis in relation to the residual tumor: grade 1 (complete regression), with an absence of histologically identifiable residual cancer and diffuse fibrosis, with or without granulomatous inflammation; grade 2, with rare residual cancer cells scattered through the fibrosis; grade 3, with an increase in the number of residual cancer cells, but fibrosis still predominating; grade 4, with residual cancer outgrowing fibrosis; grade 5 is characterized by the complete absence of regressive changes [30]. However, the prognostic value of Mandard’s TRG in cervical cancer is still undefined. In the following study, we aimed to assess the prognostic value of Mandard’s TRG in categorizing the pathological response to neoadjuvant CT/RT of locally advanced cervical cancer in order to stratify the prognosis in this subset of patients.

## 2. Materials and Methods

The study complied with the Ethical Principles for Medical Research Involving Human Subjects according to the World Medical Association Declaration of Helsinki. The study was approved by our Institutional Review Board (IRB): n° IST DIPUSVSP-17-05-2134. The clinical information was retrieved from the patient medical records and pathology reports. Patient initials or other personal identifiers did not appear in any image, and all the samples were anonymized before histologic and immunohistochemistry evaluation.

### 2.1. Study Population

From January 2011 to December 2018, 244 consecutive patients were enrolled with a diagnosis of LACC (stage IB2-IVA, FIGO staging classification, 2018) managed by CT/RT followed by completion surgery. All patients were treated at the Gynecologic Oncology Unit and Radiotherapy of the Catholic University in Rome, Italy. 

All patients had signed a written informed consent form agreeing to be submitted to all the procedures described, and for their data to be collected. 

According to current practice [31], pretreatment workup included clinical examination, abdomino-pelvic MRI, complete blood count and measurement of liver and renal function, and cystoscopy and proctoscopy if needed. Patients underwent preoperative CT/RT administered as whole pelvic irradiation in combination with cisplatin-based regimens with or without 5-fluorouracil. There were slight variations in treatment schemes of platinum-based chemotherapy or radiotherapy (total dose from 39.6 to 50.4 Gy), or upper border of the radiation field (L4–L5 vs. L3 vertebra) were used [32].

After 5/6 weeks from completion of the CT/RT stage, PET/CT and abdomino-pelvic MRI were carried out to evaluate responses to treatment depending on the RECIST criteria [33,34].

All patients after CT/RT underwent a radical hysterectomy and pelvic lymph node dissection within 6–8 weeks from the end of the therapy. Aortic lymphadenectomy was performed in case of persistence of pelvic lymph node (LN) involvement after CT/RT at imaging, intra-operative evidence of palpable, indurated or fixed pelvic and/or aortic LNs, and intra-operative assessment involved pelvic LNs at frozen section analysis. 

According to an internal protocol, patients showing a complete response or microscopic (<0.3 cm) residual started routine surveillance procedures, while patients showing macroscopic residual or involvement of pelvic and/or aortic LNs were triaged to adjuvant chemotherapy. After treatment, a follow-up was scheduled every 3 months for the first 2 years, and every 6 months thereafter. Follow-up included physical examination and CT-scan. 

### 2.2. Pathological Approach

All surgical specimens were sent to the histopathology unit and fixed in 4% formalin for 12–18 h. In the presence of macroscopic residual neoplasm, a minimum of 4–5 blocks were prepared. In the absence of macroscopic tumor, the whole cervix was dissected and embedded in paraffin.

The cervix was processed with an automatic and fully paraffin-embedded processor. As a standard procedure, at least two histological sections of 3–4 μm were cut to different levels by microtome and stained with hematoxylin and eosin stain. All surgical specimens were evaluated by an experienced gynecopathologist. Immunohistochemical stains for CK AE1/AE3, p16 and p63 were performed if necessary [35,36]. 

Morphologically, in the context of erosive cervicitis, the neoplastic cells after neoadjuvant treatment showed the following alterations: enlarged nuclei with irregular outlines, the cytoplasm could range from intensely eosinophilic to clear with vacuolation or it could have a foamy appearance; rarely, neoplastic cells could have multinucleated giant cells; the stroma could be densely fibrotic, rich in foamy histiocytes, with cholesterol clefts, necrosis and calcifications. Vessels could show myointimal thickening due to the deposit of acellular material. The endocervical non-neoplastic cells could have a reactive appearance that could simulate endocervical carcinoma, but they lack epithelial stratification, severe nuclear atypia and significant mitotic activity [18]. 

The assessment of the tumor regression grade in LACC after CT/RT was performed semi-quantitatively by using Mandard’s TRG, as follows: TRG1 = complete regression with absence of histologically identifiable residual cancer and diffuse fibrosis, with or without granulomatous inflammation; TRG2 = rare residual cancer cells scattered through the fibrosis; TRG3 = increase in the number of residual cancer cells, but fibrosis still predominating; TRG4 = residual cancer outgrowing fibrosis; TRG5 = complete absence of regressive changes. TRG 4 and 5 were joined together due to the lack of reproducibility in differentiating between the two groups [30,37,38].

### 2.3. Statistical Analysis

Patient data were recorded in an Excel database. The discrete variables were compared using the χ^2^ test; it was used to evaluate the relationship between residual disease and surgical specimens and clinic-pathologic characteristics. OS and PFS were calculated from date of diagnosis to date of death and disease progression, respectively. Patients alive and without evidence of tumor recurrence were censored at last follow-up. Patients who died without tumor recurrence were censored at the time of death. Survival was studied with Kaplan–Meier curves; significance was confirmed with the log-rank test. Mean survival ± standard deviation with 95% confidence interval (CI) and 5-year survival ± standard deviation were calculated for each prognostic group. Multivariate analysis was performed using Cox’s regression with hazard ratio (HR) calculation. SPSS Statistics 28.0 (SPSS ©) was used for the statistical calculations. In all cases, we chose a degree of significance of 95%. In the tables, we express the continuous numerical variables as the number of observed cases and mean ± standard error of the average.

## 3. Results

### 3.1. Population Characteristics

The median age of the patients was 51 years at the time of diagnosis (range 22–79). Pathological diagnoses included squamous cell carcinomas (216 cases, 88.5%) and adenocarcinomas (28 cases, 11.5%). Clinical and histological features are shown in Table 1.

The median of follow-up was 29 months in terms of PFS with a mean of 38.6 months; the median in terms of OS was 42.88 months with a mean of follow-up of 36 months. Thirty-eight patients (15.6%) died of disease and sixty-five cases (26.6%) showed recurrence of disease.

### 3.2. Four-Tiered TRG

A complete pathological response to therapy with no residual neoplastic cells (classified as TRG1) was observed in 118 patients (48.4%). In this group, mean OS was 101.6 ± 2.5 months (95% CI, 96.7 to 106.5), while 5-year OS was 94.1 ± 2.6%; mean PFS was 91.7 ± 3.6 months (95% CI, 84.7 to 98.7), with a 5-year PFS of 78.5 ± 4.5%. 

Forty-nine cases (20.1%) showed scattered residual neoplastic cells and were classified as TRG2; this group showed a mean OS of 106.1 ± 6.1 months (95% CI, 94.2 to 118.0), with a 5-year OS of 85.9 ± 6.8%; mean PFS was 84.7 ± 8.2 months (95% 68.6 to 100.8), with a 5-year PFS of 69.9 ± 7.7%.

Thirty-five cases (14.3%) were classified as TRG3 and showed a mean OS of 66.7 ± 7.5 months (95% CI, 52.1 to 81.4), with a 5-year OS of 52.1 ± 12.6%; mean PFS was 47.2 ± 5.6 months (95% CI 36.1 to 58.2), with a 5-year PFS of 42.3 ± 11.6%. 

Forty-two cases (17.2%) were classified as TRG4–5 and showed a mean OS of 53.0 ± 6.1 months (95% CI, 40.9 to 65.0), with a 5-year OS of 43.3 ± 10.3%; mean PFS was 43.0 ± 6.3 months (95% CI, 30.7 to 55.3), with a 5-year PFS of 36.3 ± 9.1%. 

Mandard’s TRG was significantly associated with both OS (*p* < 0.001) and PFS (*p* < 0.001). 

Examples of each TRG group are shown in Figure 1. Kaplan–Meier Curves for OS and PFS are shown in Figure 2.

### 3.3. Two-Tiered TRG

Observing Kaplan–Meier curves for OS, it was noted that the four curves cluster into two groups, with TRG1 and TRG2 cases showing a better outcome and TRG3 and TRG4–5 cases showing a worse outcome. 

The TRG1–2 group showed a mean OS of 109.2 ± 2.7 months (95% CI, 103.9 to 114.4), with a 5-year OS of 92.1 ± 2.6%; mean PFS was 95.5 ± 3.7 months (95% CI 88.2 to 102.7), with a 5-year PFS of 75.9 ± 4.0%. The TRG3–4-5 group showed a mean OS of 62.2 ± 5.3 months (95% CI, 51.9 to 72.6), with a 5-year OS of 47.6 ± 8.0%; mean PFS was 47.4 ± 4.7 months (95% CI, 38.2 to 56.6), with a 5-year PFS of 39.3 ± 7.3%. The two-tiered TRG was significantly associated with both OS (*p* < 0.001) and PFS (*p* < 0.001). Kaplan–Meier Curves for OS and PFS are shown in Figure 2.

Cox’s regression analysis showed that the two-tiered TRG was independently associated with both OS (HR = 5.7, 95% CI 2.5 to 13.0; *p* < 0.001) and PFS (HR = 2.8; 95% CI 1.5 to 5.2; *p* < 0.001). Among other factors, the pre-treatment FIGO stage was independently associated with OS (HR = 1.4, 95% 1.0 to 1.9; *p* = 0.046) but not with PFS (*p* = 0.087); lymphovascular space invasion and histotype were not independently associated with OS (*p* = 0.061 and 0.393, respectively) or PFS (*p* = 0.084 and 0.286, respectively). Results of the Cox regression survival analysis are shown in Table 2.

## 4. Discussion

This study showed that Mandard’s TRG stratified post-CT/RT locally advanced cervical carcinoma into prognostically significant groups, with TRG1–2 cases having better survival and TRG3–4-5 cases having a worse prognosis. The prognostic value of the two-tiered TRG is independent of pretreatment FIGO stage, lymphovascular space invasion and histotype. Having shown that Mandard’s TRG is an independent prognostic factor in post-CT/RT LACC, we assessed the importance of standardized processing of tumor resection specimens to determine histopathological regression. In addition, residual tumors in cervical cancer specimens after neoadjuvant chemotherapy appeared to be the only histopathological criterion that significantly correlated with treatment response and subsequent overall survival.

Despite a large screening program and the introduction of vaccines, cervical cancer remains a huge public health problem, especially in low- and middle-income countries. When neoadjuvant therapy is the chosen treatment in LACC, the assessment of residual disease in the pathological report after surgery is required. Moreover, a tumor regression scoring system with a significant correlation with prognosis could be an important piece of the puzzle in the era of tailored therapies [17,23,25,39,40,41,42]. Response to preoperative chemotherapy could indicate tumor chemo-sensibility, and histopathological assessment of tumor regression could correctly identify a group of patients in whom chemotherapy had a low effect. This evaluation may help to detect new therapeutic protocols to improve the clinical response and decrease morbidity [43,44]. 

Several pathological parameters have been related to prognosis in cervical cancer [45,46,47,48,49]. The pathological features required for treatment and prognostic classification are cancer stage, lymph node invasion, lymphovascular space invasion, positive surgical margins, and type. 

Our study assessed the association of prognosis with multiple pathological features, including tumor type, pre-treatment FIGO stage, and lymphovascular space invasion [50,51,52]. 

Historically, tumor types and patient outcomes have always been related. In previous studies, adenocarcinoma showed a worse prognosis in terms of OS but not of PFS [39]. The worse outcome of these patients has long been due to a relatively lower radio- and chemo-sensibility and slower regression during or after radiotherapy.

In our series, tumor histotype did not have an independent association with prognosis. However, since we collected a small case study of 23 adenocarcinomas, further studies are needed to evaluate this subset of tumors. 

Lymphovascular space invasion has shown prognostic value in early-stage cervical cancer [50,51,52], but in our series of LACC, it was independently associated with neither OS nor PFS. As is well-known in the literature, LVSI was an independent risk factor for recurrence, and pre-treatment FIGO was the only parameter that showed independent association with survival. However, in a few studies, lymphovascular space invasion was associated with a significantly decreased sensitivity to CH/RT and a poor outcome [53]. In addition, post-treatment LVSI assessment may be affected by regressive changes and be less meaningful than pre-treatment evaluation.

Recently, various histological regression scoring systems have been widely evaluated for cervical cancer, but there is currently no common standard for reporting tumor regression in resection specimens after neoadjuvant treatment [54]. Different scores have been proposed depending on the presence of residual tumors, including a five-stage score based on the amount of neoplastic cells by Takatori et al. On multivariate analysis, they showed that lymph node metastasis before NAC and anti-tumor responses were independent prognostic factors [23]. In 2008, another study introduced a three-tiered score based on the presence or absence of residual tumor and the diameter (> or <0.3 cm) of the residual tumor, but this has never been tested for prognosis [19]. Other authors evaluated pathological regression as PCR, pathological complete response (complete disappearance of tumor from cervix and lymph nodes); PR1, partial response 1 (residual disease with less than 3 mm of stromal invasion, including carcinoma in situ with or without lymphatic metastases); PR2, partial response 2 (persistent residual disease with more than 3 mm of stromal invasion). Only nodal status was found to be the strongest predictor of survival in women with LACC [55]. Gaducci et al. proposed a three-category scoring system assessed on 82 patients. Complete response refers to the complete absence of tumors in the cervix with negative nodes. Optimal partial response is defined as the presence of persistent residual disease with 3 mm stromal invasion within the cervix and negative nodes, while suboptimal response with extra-cervical residual disease is characterized by positive nodes regardless of parametria and surgical margins, and positive parametria of surgical margins with negative nodes [25]. 

To date, only one study has proposed the Mandard tumor regression grade to assess pathological regression in cervical cancer, without any correlation with prognosis [56]. 

A recent meta-analysis compared different regression scores, and of 12 studies, only the depth of residual stromal invasion was suitable for analysis, precluding the possibility of comparisons between different systems. However, the depth of residual tumor invasion was confirmed to be a reliable stratifying factor for the prognosis of LACC after neoadjuvant treatment [54]. Although further studies may enable comparison with other prognostic factors, it should be noted that our survival analysis could be limited by the lack of deep stromal invasion.

Comparing the response rate with the literature is difficult due to limited data on surgical strategies following neoadjuvant treatment for locally advanced cervical cancer. In other solid tumors, the regression system proposed by Mandard is widely used, and pathologists have great experience in evaluating it and are more familiar with its clinical implications. 

For example, the Mandard score is widely used in patients with gastroesophageal and rectal cancer that undergo surgical resection after neoadjuvant therapy. In this subgroup of patients, different regression systems could be used to evaluate the response to therapy which could be divided into tumor regression systems that refer to the amount of fibrosis and tumor, such as Mandard and Dworak scores, or that use the percentage of the tumor as a reference for evaluating regression, such as the Becker and the Rödel system [57,58,59,60,61]. 

According to the treatment response expressed in terms of residual tumor tissue and the degree of tumor regression, TRG may provide more informative results than solely relying on the pT category. Although some studies have questioned the prognostic value of these scores, mainly because of the lack of long-term outcomes in most of these patients, current studies have established an index that seems to predict outcomes based on the tumor regression score in breast and gastrointestinal pathology [57]. 

Indeed, in our study, we showed that Mandard’s TRG is significantly associated with both OS and PFS in post-CT/RT LACC. In fact, mean OS dropped from 101.6 months for TRG1 to 53 months for TRG3–4. Interestingly, Kaplan–Meier curves highlighted that TRG2 cases had a mean OS (106.1 months) similar to TRG1 cases, while the mean OS of TRG3 cases (66.7 months) was significantly decreased. A comparable trend was observed in the PFS analysis, with a similar mean PFS between TRG3 and TRG4–5 cases (47.2 and 43 months, respectively), while the mean PFS of TRG2 cases was nearer to that of TRG1 cases (84.7 and 91.7 months, respectively). These findings prompted us to test the prognostic value of a dichotomized TRG (TRG1–2 vs. TRG3–4-5). We found that the dichotomized TRG was significantly associated with both OS and PFS independently from other factors such as pre-treatment FIGO stage, tumor histotype, and lymphovascular space invasion. The dichotomized TRG was also the only factor independently associated with both OS and PFS, differentiating “good responders” from “bad responders”. The strength of a dichotomized TRG may lie not only in a stronger association with survival, but also in an easier applicability. In fact, the distinction between TRG3, 4 and 5 depends on the ratio between residual tumor and regressive fibrosis. 

Considering the histological assessment, many challenges have been encountered in clinical practice. In our study, we applied a score based on the induced tissue fibrosis compared with the residual tumor; each tier corresponds to different levels of fibrosis and regressive changes. Cervical stroma are composed of densely packed collagenous fibers, which can be difficult to differentiate from regressive fibrosis. In particular, the evaluation of preponderant tumor components could be challenging, leading to confusion between minimal cervical wall fibrosis and minimal regression. The presence of some resorptive changes made the fibrosis assessment easier. For example, foamy histocytes, cholesterol clefts, foreign body reaction, necrosis and calcifications were specific for therapy-induced regression [18,59]. In addition, acellular mucin and mucinous changes can be observed. However, we also had to consider that the post-treatment stroma could show significant changes, as vascular proliferation, atypical reactive endothelial cells, thrombi, and bizarre stromal fibroblasts. These alterations affect both tumoral and non-tumoral stroma and should not be misunderstood. In addition, poor reproducibility and difficulty in scoring may result from differentiating between TRG3 and TRG4 based on the amount of fibrosis. The modified Mandard system allows us to overcome this problem, showing by far the highest usability.

In the literature, another critical issue of the significance of TRG score is a variety in treatment protocols and different time frames between surgery and therapy. Radiotherapy dose and schedule and combination with chemotherapy could determine different pathological regression, but in our study, all patients were treated with the same protocol and comparable time setting.

Moreover, several studies and meta-analyses have provided evidence supporting the role of TRG as a crucial predictor of survival, notably in gastroenteric pathology. A strong correlation between TRG and patient outcomes has been observed in upper gastrointestinal cancers, where a deficiency in tumor response corresponds to a worse outcome. Based on these studies, we expect that in cervical cancer the prognostic value of TRG will be valid in addition to the classical staging system and be adopted in pathological reporting.

Finally, tumor regression not only represents prognostic value, but could indicate an early response to therapy in a therapeutic context and potentially allows protocols to be adapted according to the tumor response. 

On the other hand, data on the impact of TRG in gynecologic pathology are currently lacking. The evaluation of tumor regression could become pivotal for clinical decision making, surgical approaches, and post-operative adjuvant treatment.

In addition, the impact of immune checkpoint inhibitors and other treatments besides conventional chemo- and radiotherapy on the tumor is poorly understood and could be an interesting field in the future. Tissue examination could provide information on the effects of new drugs and their resistance.

## 5. Conclusions

Although we hope that in the near future, the incidence of cervical cancer will decrease thanks to large vaccine programs and screening protocols, for the next few years we have to face this tumor, especially in advanced stages. 

The present research demonstrated that Mandard’s TRG is a significant and independent prognostic factor in post-CT/RT locally advanced cervical carcinoma. Our results showed that Mandard’s TRG might be dichotomized as TRG1–2 vs. TRG3–4–5, possibly improving the applicability of the score. 

We suggest that a dichotomized Mandard’s TRG could be used to stratify prognosis in post-CT/RT LACC, with potential benefits in defining post-treatment adjuvant therapy. Further studies are encouraged in this field.

## Figures and Tables

**Figure 1 diagnostics-13-03228-f001:**
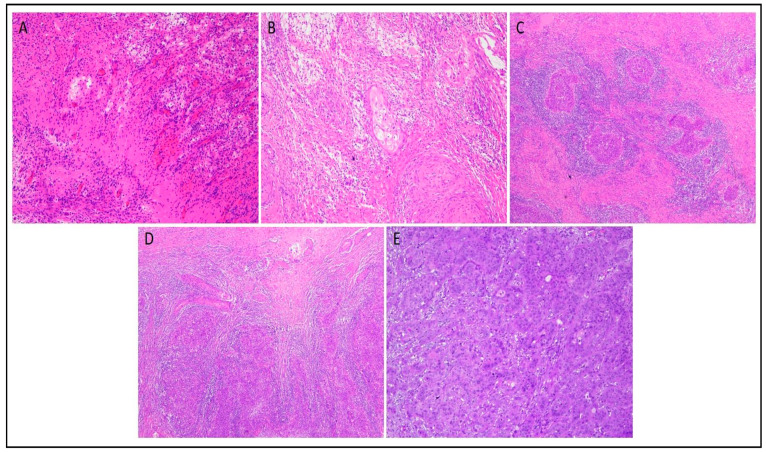
Tumor regression grading (TRG) after preoperative chemo-radiotherapy in cervical cancer: TRG1 (HE stain 10×) (**A**); TRG2 (HE stain 10×) (**B**); TRG3 (HE stain 10×) (**C**); TRG4 (HE stain 10×) (**D**) and TRG 5 (HE stain 20×) (**E**).

**Figure 2 diagnostics-13-03228-f002:**
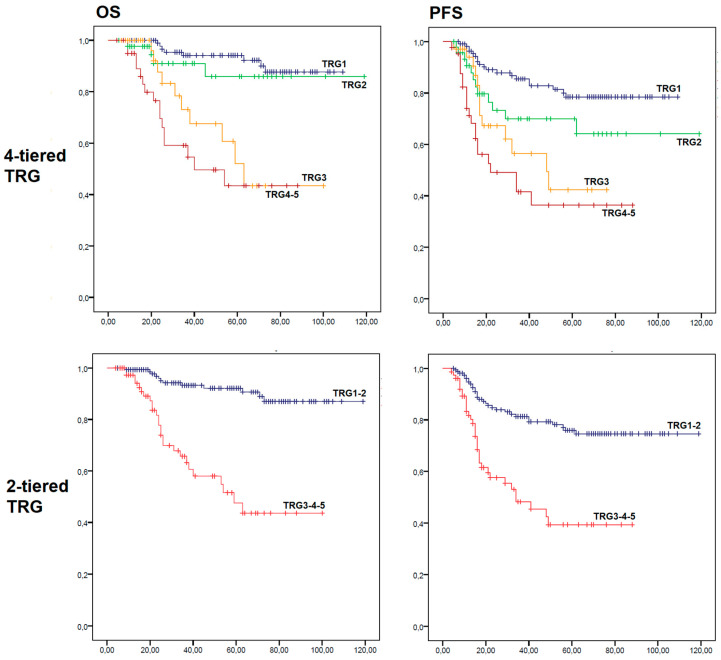
Kaplan–Meier disease free and overall survivalurves for OS and PFS. Tumor regression grade (TRG).

**Table 1 diagnostics-13-03228-t001:** Clinical and pathological data.

Variables	
**Age, years**	
Median (range)	51 (22–79)
**Histotype**	
Squamous cell carcinoma	216 (88.5%)
Adenocarcinoma	23 (9.2%)
Adenosquamous carcinoma	5 (2.3%)
**FIGO stage**	
IB3-IIB	130 (53.2%)
IIIA-IIIB	17 (7%)
IIIC1	89 (36.5%)
IIIC2	6 (2.5%)
IVA	2 (0.8%)
**Lymph vascular space invasion**	
Absent	186 (76.2%)
Present	58 (23.8%)
**Mandard’s tumor regression grade (TRG)**	
TRG1 (no residual)	118 (48.4%)
TRG2 (scattered cells only)	49 (20.1%)
TRG3 (residual < fibrosis)	35 (14.3%)
TRG4–5 (residual > fibrosis)	42 (17.2%)

**Table 2 diagnostics-13-03228-t002:** Results of Cox regression survival analysis.

Variable	Overall Survival		Progression-Free Survival	
	HR (95% CI)	*p*-Value	HR (95% CI)	*p*-Value
Two-tiered TRG	5.7 (2.5–13.0)	<0.001	2.8 (1.5–5.2)	<0.001
Pre-treatment FIGO stage	1.4 (1.0–1.9)	0.046	1.2 (0.9–1.5)	0.087
Lymphovascular space invasion	2.0 (0.9–4.3)	0.061	1.7 (0.9–3.2)	0.084
Histotype (squamous cell vs. others)	0.7 (0.2–1.7)	0.393	0.6 (0.3–1.5)	0.286

## Data Availability

Data available on request due to restrictions, e.g., privacy or ethical.

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
