# Peer review of "Prognostic Value of Mandard’s Tumor Regression Grade (TRG) in Post Chemo-Radiotherapy Cervical Cancer"

_diagnostics, 2023, doi:10.3390/diagnostics13203228_

Round 1
Reviewer 1 Report
•Dear Professor,I have read the papers by the team of professors Gabriella Ferrandina and Gian Franco Zannoni. I am very interested in them. The current study represents a pioneering investigation on the role of TRG in forecasting the outcome of LACC. It is crucial for future research.
I have some questions regarding the content of the manuscript. •1. Was lymphovascular space invasion discovered in the surgical tissue following CT/RT? •2. What is the difference between regressive fibrosis with dense collagenous fibers in nature? •3. Was there the difficulty in distinguishing between TRG3, 4, and 5 in other tumors, such as rectal cancer, soft tissue sarcoma? •How can the regression grade of metastatic lymph nodes be determined?
Author Response
I have some questions regarding the content of the manuscript.
- 1. Was lymphovascular space invasion discovered in the surgical tissue following CT/RT?
Yes, the lymphovascular space involvement was evaluated in surgical resection after CT/RT but we did not found a strong correlation with the prognosis.
- 2. What is the difference between regressive fibrosis with dense collagenous fibers in nature?
Morphologically it is a very difficulties challenge in daily practice routine. We generally used the association with other features, for example we use the presence of myxoid change in the fibrosis, the presence of foamy hystiocytes and calcification, in order to define a fibrosis related to the neoadjuvant treatment.
- 3. Was there the difficulty in distinguishing between TRG3, 4, and 5 in other tumors, such as rectal cancer, soft tissue sarcoma?
Yes, it is a common issue among pathologist, indeed we suggest to split the Mandard-TRG in 2 groups TRG1-2 and TRG3,4,5 according also to the clinical outcome.
4 •How can the regression grade of metastatic lymph nodes be determined?
In this study we did not assed it because of the low number of cases with lymph-nodes involvement. However, in practice, we use Mandard score.
Reviewer 2 Report
The manuscript "Prognostic Value of Mandard’s Tumor Regression Grade (TRG) in Post Chemo-Radiotherapy Cervical Cancer" is an interesting manuscript on the prognostic value of Mandard’s TRG in categorizing the pathological response of LACC to neoadjuvant CT/RT. The work is complete and well structured, giving important lessons to clinicians and scientific literature. The design of the project is appropriate and the images are satisfactory. The language is acceptable. It represents a valid work and it gives the opportunity to focus attention on possible future studies on this field.
Author Response
Thank you very much.